

# Cloud Condensation Nuclei (CCN) Activity Analysis of Low-hygroscopicity Aerosols Using the Aerodynamic Aerosol Classifier (AAC)

Kanishk Gohil[1] and Akua Asa-Awuku[1,2]

[1]Department of Chemical and Biomolecular Engineering, University of Maryland, College Park, MD 20742, United States
[2]Department of Chemistry and Biochemistry, University of Maryland, College Park, MD 20742, United States

**Correspondence:** Akua Asa-Awuku (asaawuku@umd.edu)

**Abstract.** The Aerodynamic Aerosol Classifier (AAC) is a novel instrument that size-selects aerosol particles based on their mechanical mobility. So far, the application of an AAC for Cloud Condensation Nuclei (CCN) activity analysis of aerosols has yet to be explored. Traditionally, a Differential Mobility Analyzer (DMA) is used for aerosol classification in a CCN experimental setup. A DMA classifies particles based on their electrical mobility. Substituting the DMA with an AAC can

eliminate multiple charging artefacts as classification using an AAC does not require particle charging. In this work, we describe an AAC-based CCN experimental setup and CCN analysis method. We also discuss and develop equations to quantify the uncertainties associated with aerosol particle sizing. To do so, we extend the AAC transfer function analysis and calculate the measurement uncertainties of the aerodynamic diameter from the resolution of the AAC. The analyses framework has been packaged into a Python-based CCN Analysis Tool (PyCAT 1.0) open source code, which is available on GitHub for public use.

Results show that the AAC size-selects robustly (AAC resolution is 10.1, diffusion losses are minimal and particle transmission is high) at larger aerodynamic diameters ($\geq \sim$85nm). The size-resolved activation ratio is ideally sigmoidal since no charge corrections are required. Moreover, the uncertainties in the critical particle aerodynamic diameter at a given supersaturation can propagate through droplet activation and the subsequent uncertainties with respect to the single-hygroscopicity parameter ($\kappa$) are reported. For a known aerosol such as sucrose, the $\kappa$ derived from the critical dry aerodynamic diameter can be up to $\sim$50%

different from the theoretical $\kappa$. In this work, we do additional measurements to obtain dynamic shape factor information and convert the sucrose aerodynamic to volume equivalent diameter. The volume equivalent diameter applied to $\kappa$- Köhler theory improves the agreement between measured and theoretical $\kappa$. Given the limitations of the coupled AAC-CCN experimental setup, this setup is best used for low hygroscopicity aerosol ($\kappa \leq 0.2$) CCN measurements.

## 1 Introduction

Cloud Condensation Nuclei (CCN) activity is defined as the ability of an aerosol particle to facilitate the condensation of water vapor on its surface; the condensation occurs in supersaturated ambient conditions resulting in the formation of droplets. The use of size-resolved aerosol number concentrations obtained with the help of counting instruments is a reliable method for determining the CCN activity of aerosols (e.g. but not limited to Petters et al. (2007), Rose et al. (2008), Moore et al. (2010),





Vu et al. (2015), Zieger et al. (2017), Barati et al. (2019)). Currently, the most common method for studying CCN activation

uses a CCN counter (CCNC) and couples it with an aerosol classifier. CCN activity measurements have consistently improved over the past few years since the development and commercialization of the Continuous-Flow Streamwise Thermal Gradient CCN Chamber (CFSTGC) developed by the Droplet Measurement Technologies (DMT) (Roberts and Nenes (2005), Rose et al. (2008), Lathem and Nenes (2011)) and it is widely used. However, there are several commercially available options to size-select ultrafine particles.

An aerosol classifier size-selects and generates a monodisperse aerosol from a polydisperse aerosol population. The most widely used aerosol classifier for CCN measurements is the Differential Mobility Analyzer (DMA) (Knutson and Whitby (1975), Rader and McMurry (1986), Wang and Flagan (1990)). The DMA classifies the aerosol particles based on their electrical mobility; a charge distribution is applied on the particles which then pass through an external electrostatic field that is generated by varying the voltage difference across the DMA column. Many CCN studies use the DMA in "scanning mode"

for which stepwise voltage is applied across the aerosol flow to generate monodisperse particles between ∼10-500nm. The size-selected particles are then counted by a Condensation Particle Counter (CPC) and a parallel CCNC to obtain the number size distributions for the total aerosol particles (Condensation Nuclei, CN) and activated droplets (CCN) respectively, at a constant instrument supersaturation. The aerosol CN and CCN number size distributions are then combined to calculate the size-resolved activation ratio $\left(\frac{CCN}{CN}\right)$ of the aerosol at the given instrument supersaturation.

A major limitation of this method is associated with the working mechanism of the DMA. The DMA uses a neutralizer (e.g., Kr-85, soft X-ray, or Po-210) to distribute electric charge to classify the polydisperse particles. The particles may receive multiple unit charges depending on the charging efficiency of the neutralizer. As a result, the particles carrying a unit charge possess the same electrical mobility as larger particles carrying a higher integral charge. Therefore, the perceived monodisperse aerosols likely contain a mixture of different-sized particles. This issue is known and can lead to discrepancies in the size-

resolved activation ratio $\left(\frac{CCN}{CN}\right)$ (Moore et al. (2010)). Hence, charge correction algorithms (Gunn (1956), Fuchs (1963), Wiedensohler (1988)) are commonly applied to resolve particle multiple charging issues and data correction is applied in CCN software. Multiple charging errors can still affect the reliability and efficacy of CCN activation data.

The multiple charging issues in electrical mobility-based classifiers have led to the development of instruments that use particle mechanical mobility. Classifiers can measure the relaxation time in pressurized flow or free-molecular (vacuum)

regimes (e.g. but not limited to Conner (1966), Marple et al. (1991), Keskinen et al. (1992), Chein and Lundgren (1993), Flagan (2004)). Recently, the working principle and instrumentation details for an Aerodynamic Aerosol Classifier (AAC) were described (Tavakoli and Olfert (2013), Tavakoli et al. (2014)). The AAC does not require particle charging for size-selection and does not produce multiple charging artifacts (Yao et al. (2020)). The AAC classifies particles with respect to their relaxation time, and reports the aerodynamic diameter.

The AAC has been used with different instruments. Johnson et al. (2018) used the AAC in tandem with the Scanning Mobility Particle Sizer (SMPS) to characterize the transfer function of the AAC. The AAC can classify particles as large as $6\mu$m (Johnson et al. (2018)). Furthermore, the AAC in tandem with a DMA can determine the aerosol dynamic shape factor (Tavakoli and Olfert (2014), Barati et al. (2019), Yao et al. (2020), Tran et al. (2020)) and particle effective density (Tavakoli





and Olfert (2014), Peng et al. (2021b)). Sang-Nourpour and Olfert (2019) and Tran et al. (2020) discuss methods for Optical

Particle Counter (OPC) calibration using an AAC.

In short, the AAC is increasing in popularity (e.g. but not limited to Johnson et al. (2020), Su et al. (2021), Johnson et al. (2021)). However, the scientific knowledge of coupling an AAC with a CCNC is limited. One previous study (Barati et al. (2019)) published results for the CCN analysis of low-hygroscopicity aerosols but did not investigate the uncertainties in AAC-CCN size resolved measurements and CCN activity predictions. To our knowledge, the validation of AAC-CCNC coupling

on CCN measurement and prediction has not been studied before, and hence the AAC-CCNC coupled system is currently not well understood. This work explains the AAC-CCNC coupling for CCN activity measurements and uncertainties associated with size-selection, number size distributions and CCN activity estimates employing the AAC transfer function.

In addition to a standardized experimental protocol for an AAC-CCNC setup, a computational tool also needs to be developed for CCN analysis. Currently, the Scanning Mobility CCN Analysis (SMCA) (Moore et al. (2010)) package is widely

used to calculate the CCN activity of aerosols using their electrical mobility-classified number size distribution data. The processed size-distribution data from the SMCA can be analyzed using the Köhler theory (Köhler (1936), Seinfeld and Pandis (2016)). SMCA has been shown to efficiently perform functions that include inversion of time series measurements to obtain size-resolved data (Wang and Flagan (1990)), and multiple charge correction using the algorithm given by Wiedensohler (1988). SMCA works well for a variety of organic and inorganic aerosols to estimate their CCN activity (e.g. but not limited

to Moore et al. (2010), Padró et al. (2012), Giordano et al. (2015), Fofie et al. (2018), Barati et al. (2019), Vu et al. (2019), Dawson et al. (2020), Peng et al. (2021a)). So far, there is no computational analysis tool for data processing or CCN analysis using their aerodynamic measurements based on AAC-CCNC setup.

In this work we couple the AAC with the CCNC, ascribing to the aforementioned advantages and novelty of the AAC, for CCN activity analysis. We develop and test an experimental setup and CCN analysis tool. The analysis tool was developed

in Python (PyCAT 1.0, described in Section 2.3) and is available on GitHub for public use. In the following sections, we first describe the experimental setup to size select and count particles. We then describe the theory and mathematical formulations used in CCN analysis of aerosols. After that, we discuss the uncertainties associated with aerodynamic size selection and the propagated error into the CCN activity analysis, as well as the impact on the subsequently derived single-hygroscopicity parameter ($\kappa$) values.

## 2  Experimental Design and Methodology

### 2.1  Instruments and Setup

A Cambustion™ Aerodynamic Aerosol Classifier (AAC) size-selected polydisperse aerosol. Briefly described here, the AAC contains 2 concentric cylindrical columns for particle selection. The schematic of a typical AAC is shown in Figure 1. The particles are introduced into the AAC from inside the inner cylinder and the aerosol flow is then passed into the space between

the 2 cylinders. The particles move with axial and radial velocities because of the rotation of the cylinders. The rotational speed steps across a range of values when the AAC is operated in "scanning mode". Each of the rotational speeds correspond





to a relaxation time and aerodynamic diameter. At different speeds, the particles can hit the inner surface of the outer cylinder depending on their size. The outer cylinder has an opening through which the particles of an optimum size corresponding to a specific rotational speed can pass through. Particles larger than the threshold optimum size hit the cylindrical surface before

the opening, and the ones smaller than the threshold, exit the classifier along with the exhaust flow. The working principle of the AAC has been described previously in extensive detail (Tavakoli and Olfert (2013), Tavakoli et al. (2014), Johnson et al. (2018)).

Figure 2 shows the experimental setup used in this study. The classified aerosol was split into 2 streams - the first stream was passed through a Condensation Particle Counter (CPC, TSI 3776) to obtain total aerosol particle counts (condensation

nuclei, CN), and the second stream was passed through a DMT Continuous-Flow Streamwise Thermal-Gradient CCN Chamber (CFSTGC, or simply CCNC; Roberts and Nenes (2005)) to obtain activated aerosol particle counts (cloud condensation nuclei, CCN). The CCNC consists of a cylindrical chamber that has internally wetted walls to maintain an approximately constant supersaturation along the CCNC column. A series of experiments were performed with sucrose at different instrument supersaturations (between 0.2% and 0.6%). Sucrose is a highly water-soluble, moderately hygroscopic oligomer that

is an atmospherically relevant aerosol from biogenic sources (Dawson et al. (2020)). The CCN properties of sucrose have been well-studied and characterized (e.g. but not limited to Rosenørn et al. (2006), Petters and Kreidenweis (2007), Xu et al. (2014), Wang et al. (2017), Dawson et al. (2020)). Sucrose was selected as an appropriate choice of aerosol to benchmark the AAC-CCNC setup.

The polydisperse aerosol population was generated from an aqueous solution using a Collison atomizer. The aerosol

was passed through a series of 2 diffusion driers (for drying to <10% RH) and then introduced into the AAC to generate monodisperse aerosol. The atomization method typically produces dry particles in the submicron size range. A total sample flow rate of 0.8 L min$^{-1}$ was split between 0.3 and 0.5 L min$^{-1}$ for the CN and CCN measurements, respectively. Additionally, a sheath flow rate of 8 L min$^{-1}$ was applied to maintain a sheath-to-sample flow ratio of 10:1. Furthermore, the AAC was maintained at a working temperature and pressure of 21.5°C and 1 atm, respectively. The CCNC instrument supersaturations

were calibrated using ammonium sulphate ($(NH_4)_2SO_4$) (Rose et al. (2008)). The details of CCNC calibration performed using DMA-based size-resolved $(NH_4)_2SO_4$ measurements from 0.2 to 0.6% supersaturation are provided in supplemental information (S1).

The AAC was operated in the "step-scanning mode". In step-scanning mode, there is a transit time and stabilization (delay) time when the AAC advances from one rotational speed setpoint to another. Each rotational speed is related with a

corresponding size bin, and here we ran the AAC between successive size bins for 14.5 seconds (transit time of 9.5 seconds and delay time of 5 seconds). Increasing the stabilization interval improves the repeatability of the particle counts and reduces uncertainties due to particle diffusion at lower sizes. The measured CCN to CN activation ratio ($\frac{CCN}{CN}$) was calculated for each size-selected aerodynamic diameter. A sigmoidal fit was applied to the size-resolved activation ratio. The critical dry diameter is defined at the 50% activation efficiency at a constant instrument supersaturation and was reported every 30 minutes and

repeated 5 times for the AAC-CCN experimental setup.





## 2.2 CCN Activation Theory

The critical dry diameter and instrument supersaturation can be used in Köhler theory (Köhler (1936), Seinfeld and Pandis (2016)) to estimate the size-independent single-hygroscopicity parameter ($\kappa$) of the aerosol species. $\kappa$ of an aerosol species is calculated as follows (Petters and Kreidenweis (2007)),

$$\kappa = \frac{4A^3}{27D_{p_{50}}^3 ln^2(S)}; \text{where } A = \frac{4M_w\sigma_s}{RT\rho_w} \tag{1}$$

In the above expression, $D_{p_{50}}$ is the critical dry diameter of the aerosol species at supersaturation $S$. Physically, $D_{p_{50}}$ is a threshold size for activation; particles larger than this threshold are assumed to fully activate and convert into droplets and those smaller than the threshold remain unactivated. $M_w$, $\sigma_w$ and $\rho_w$ correspond to the molar mass, surface tension and density of water, respectively. $R$ is the universal gas constant, and $T$ is the average temperature inside the CCNC column. Under the Köhler theory framework, the $\kappa$ of an aerosol species can be related to the molar mass ($M_s$), density ($\rho_s$), and Van't Hoff factor ($\nu_s$) of the solute (Sullivan et al. (2009), Mikhailov et al. (2013)),

$$\kappa = \frac{\nu_s\rho_s M_w}{\rho_w M_s} \tag{2}$$

Eq. 2 assumes complete aqueous solubility of the aerosol species. Past studies have found sucrose $\kappa$ from CCN measurements (obtained from Eq. 1) in the range of 0.06-0.1 (e.g. but not limited to Xu et al. (2014), Wang et al. (2015), Ruehl et al. (2016), Wang et al. (2017), Dawson et al. (2020)). Furthermore, the theoretical $\kappa$ of sucrose (obtained from Eq. 2) is 0.084, and implies that the previously reported $\kappa$ estimates of sucrose are in good agreement with the theoretical $\kappa$ of sucrose. Therefore, the theoretical $\kappa$ (from Eq. 2) can also be used to validate the sucrose $\kappa$ derived from the AAC-CCNC setup.

## 2.3 Python-based CCN Analysis Toolkit (PyCAT 1.0)

Each step-scanning mode timeseries using the AAC-CCNC setup measures 90 CN datapoints and 1400 CCN datapoints. Therefore, a computationally efficient method is required to synchronize and analyze the AAC and CCNC datasets. A computer code (Python-based CCN Analysis Toolkit, PyCAT) was developed to analyze both SMPS and AAC size-resolved CCN data for CCN activity analysis. The code is written in Python3.7 and uses the most recent version of the built-in libraries. It can perform timeseries data synchronization and analysis, CCN activity analysis (section 2.2) and uncertainty analysis (section 3). In addition, the code provides aerosol sizing properties at the point of activation and Köhler theory analysis based on user inputs. Additionally, the code is flexible and allows the user to organize and visualize the post analysis data. An open-source code has been completely packaged with the necessary capabilities and is available on GitHub for public use. Here we demonstrate the application of PyCAT for the first time. We use PyCAT for CCN activity analysis and to quantify the uncertainties associated with aerodynamic measurements and how they manifest in the CCN activity.





## 3 Uncertainty Analysis of Measurements

The uncertainty analysis for particle size-selection using the AAC in step-scanning mode has been described in detail previously (Johnson et al. (2018), Yao et al. (2020)). In this section, we briefly describe the derivation of AAC uncertainty and fully describe the effects of size-selection for CCN activity and single-parameter hygroscopicity uncertainty analysis.

Aerosol particles moving with an axial speed $\nu$ through the AAC column experience drag force. The drag force on a particle of an assumed spherical shape can be expressed as,

$$F_{drag} = \frac{\nu}{B_{spherical}} \tag{3}$$

where $B_{spherical}$ is defined as the mechanical mobility of the spherical particle (Tavakoli and Olfert (2014), Johnson et al. (2018), Yao et al. (2020)). For a given set of AAC operating conditions, $B_{spherical}$ can be determined as (Tavakoli and Olfert (2014), Yao et al. (2020)),

$$B_{spherical} = \frac{C_c(d_{spherical})}{3\pi\mu d_{spherical}} \tag{4}$$

where $\mu$ is the dynamic viscosity of the surrounding gas, $d_{spherical}$ is the particle diameter under the assumptions of sphericity, and $C_c(d_{spherical})$ is the Cunningham's slip correction factor of the particle with the diameter $d_{spherical}$ (described in supplemental section S2).

The particle drag force is balanced by the particle centrifugal force in the AAC for size-selection (Tavakoli and Olfert (2013)). The particle centrifugal force is defined as follows,

$$F_{centrifugal} = m\omega^2 r \tag{5}$$

where $m$, $\omega$ and $r$ are the mass, rotational speed, and radial position of the particle, respectively. The aerosol particle relaxation time, $\tau = mB_{spherical}$. Using this definition, the force balance expression $\tau$ is expressed as,

$$\tau = \frac{\nu}{\omega^2 r} \tag{6}$$

The maximum particle relaxation time ($\tau^*$) is calculated as follows (Tavakoli et al. (2014)),

$$\tau^* = \frac{Q_{sh} + Q_{exh}}{\pi\omega^2(r_1 + r_2)^2 L} \tag{7}$$

where $r_1$, $r_2$ and $L$ denote the classifier inner radius, outer radius, and length respectively. $Q_{sh}$ and $Q_{exh}$ are the inlet sheath flow and outlet exhaust flow, respectively. In this study, $Q_{sh}$ and $Q_{exh}$ were fixed by the CPC sample flowrate. $\omega$ is the only variable parameter in Eq. 7, and defines the setpoint for size-selection and determines the $\tau^*$ corresponding to particles of desired aerodynamic diameter.

The particle relaxation time can also be expressed in terms of the particle aerodynamic diameter as follows (Johnson et al. (2018)),

$$\tau = \frac{C_c(d_{ae})\rho_0 d_{ae}^2}{18\mu} \tag{8}$$


where $\rho_0$ is the reference density of 1000 kg/m$^3$ and $C_c(d_{ae})$ is the Cunningham slip correction factor of the particle with aerodynamic diameter $d_{ae}$. The aerodynamic diameter of a particle is defined for a spherical particle with a density of 1000kg/m$^3$.

A non-dimensional relaxation time, $\widetilde{\tau} = \frac{\tau}{\tau^*}$ is calculated by dividing Eq. 8 with 7.

Previous studies have developed models to calculate the probability of selecting a particle passing through the AAC, otherwise known as the AAC transfer function (TF) (Tavakoli and Olfert (2013), Johnson et al. (2018)). Tavakoli and Olfert (2013) developed the AAC transfer function following the methodology from Knutson and Whitby (1975) and Stolzenburg (1989). In this work, the non-diffusing particle streamline TF theory is used to describe particle classification (Tavakoli and Olfert (2013)). The AAC TF is denoted by $\Omega$, and for ideal non-diffusion conditions, it is defined as follows (Martinsson et al. (2001), Tavakoli and Olfert (2013), Johnson et al. (2018)),

$$\Omega_{ND}(\widetilde{\tau},\beta,\delta) = \frac{1}{2\beta(1-\delta)} \cdot \left[|\widetilde{\tau} - (1+\beta)| + |\widetilde{\tau} - (1-\beta)| - |\widetilde{\tau} - (1+\beta\delta)| - |\widetilde{\tau} - (1-\beta\delta)|\right] \tag{9}$$

where $\beta = \frac{(Q_s + Q_a)}{(Q_{sh} + Q_{exh})}$ and $\delta = \frac{(Q_s - Q_a)}{(Q_s + Q_a)}$, such that $Q_a$ is the inlet aerosol flow, and $Q_s$ is the outlet sample flow. The AAC was operated under balanced flow conditions ($Q_s = Q_a$ and $Q_{sh} = Q_{exh}$), and thus $\beta$ and $\delta$ were reduced to $\frac{Q_s}{Q_{sh}}$ and 0, respectively. Under the balanced flow assumption, Eq. 9 can be simplified to (Johnson et al. (2018)),

$$\Omega_{ND,B}(\widetilde{\tau},\beta) = \frac{1}{2\beta} \cdot \left[|\widetilde{\tau} - (1+\beta)| + |\widetilde{\tau} - (1-\beta)| - 2 \cdot |\widetilde{\tau} - 1|\right] \tag{10}$$

The non-ideal particle behavior was accounted for by incorporating a transmission efficiency ($\lambda_\Omega$) and transfer function width factor ($\mu_\Omega$) in the TF (described in supplemental section S2). The resulting TF for non-ideal, non-diffusing, balanced flow conditions is expressed as (Johnson et al. (2018)),

$$\Omega_{ND,B,NI}(\widetilde{\tau},\beta,\lambda_\Omega,\mu_\Omega) = \frac{\lambda_\Omega \cdot \mu_\Omega^2}{2\beta} \cdot \left[\left|\widetilde{\tau} - \left(1 + \frac{\beta}{\mu_\Omega}\right)\right| + \left|\widetilde{\tau} - \left(1 - \frac{\beta}{\mu_\Omega}\right)\right| - 2 \cdot |\widetilde{\tau} - 1|\right] \tag{11}$$

Figure 3 compares the theoretical TFs for ideal (Eq. 10) and non-ideal (Eq. 11) particle behaviors under the balanced flow, non-diffusion AAC framework. The two transfer functions are shown for a particle aerodynamic diameter of 150 nm ($\tau$=147.7ns, Eq. 8).

The AAC resolution can be determined from the TF broadening relative to the setpoint at $\tau = \tau^*$ (or, $\widetilde{\tau} = 1$). The AAC resolution can be correlated with the uncertainty associated with the relaxation time or aerodynamic diameter. Particles classified by the AAC only contain a narrow range of aerodynamic diameters. The AAC resolution is expressed as $\frac{1}{R_\tau} = \frac{\Delta\tau}{\tau} = \frac{Qs}{Qsh}$ and assumes the flows to be balanced, laminar and constant (Yao et al. (2020)). The AAC resolution can also be expressed in the coordinates of the aerodynamic diameter as $\frac{1}{R_{ae}} = \frac{\Delta dae}{dae}$ which forms the basis to determine the uncertainties associated with the aerodynamic diameters. Using Eq. 11, the uncertainty in relaxation time is (Yao et al. (2020)),

$$\frac{\Delta\tau}{\tau} = \frac{\delta Q_{sh}}{Q_{sh}} - 2\frac{\delta\omega}{\omega} - 2\frac{\delta r}{r} - \frac{\delta L}{L} \tag{12}$$

which can further be used to derive the uncertainty associated with the corresponding aerodynamic diameter as follows,

$$\frac{\Delta d_{ae}}{d_{ae}} = \frac{\Delta\tau}{\tau} \cdot \left[\frac{d_{ae} + \alpha_c \cdot \lambda + \beta_c \cdot \lambda \cdot e^{\left(-\gamma_c \cdot \frac{d_{ae}}{\lambda}\right)}}{2d_{ae} + \alpha_c \cdot \lambda + \beta_c \cdot \lambda \cdot e^{\left(-\gamma_c \cdot \frac{d_{ae}}{\lambda}\right)} \cdot \left(1 - \gamma_c \frac{d_{ae}}{\lambda}\right)}\right] \tag{13}$$





where $\alpha_c = 2.33$, $\beta_c = 0.966$ and $\gamma_c = 0.4985$ are the Cunningham slip correction factor coefficients taken from Kim et al. (2005), and $\lambda$ is the mean free path of the particles. The aerodynamic diameter can be converted to the volume equivalent

diameter, which is a more accurate representation of the particle morphology and size. The volume equivalent diameter is expressed using the dynamic shape factor and aerosol density as follows,

$$d_{ve} = d_{ae}\sqrt{\frac{\chi \rho_0 C_c(d_{ae})}{\rho_p C_c(d_{ve})}} \tag{14}$$

where $\rho_p$ is the particle density, $C_c(d_{ve})$ is the Cunningham's slip correction factor for the particle with the volme equivalent diameter, $d_{ve}$, and $\chi$ is the size-dependent dynamic shape factor. The uncertainties in the volume equivalent diameter are

quantified using the uncertainties in measured aerodynamic diameter and dynamic shape factor as follows,

$$\frac{\Delta d_{ve}}{d_{ve}} = \frac{\Delta d_{ae}}{d_{ae}} + \frac{1}{2\chi\rho_0}\frac{\Delta C_c(d_{ae})}{C_c(d_{ae})} - \frac{1}{2\rho_p}\frac{\Delta C_c(d_{ve})}{C_c(d_{ve})} + \frac{1}{2}\frac{\Delta\chi}{\chi} \tag{15}$$

The uncertainty given by Eq. 15 has a direct implication to the aerosol $\kappa$. For a given supersaturation, the uncertainty in $\kappa$ is dependent on the uncertainty in the volume equivalent diameter, and is expressed as,

$$\frac{\Delta\kappa}{\kappa} = -3 \cdot \frac{\Delta D_{p50}}{D_{p50}} \tag{16}$$

Eq. 16 implies that the relative uncertainty in $\kappa$ is theoretically 3 times more than that of the critical dry diameter. In Eq. 16, the $D_{p50}$ can either be the critical dry electrical mobility, aerodynamic or volume equivalent diameter. In this work, the uncertainties in Eq. 16 are evaluated with respect to volume equivalent diameters derived from the measured electrical mobility and aerodynamic diameters. Another important point to note here, is that since the activation diameter varies with supersaturation, the uncertainty at every activation diameter will also be different. This implies that for each measured activation diameter, the

uncertainty in aerosol $\kappa$ will vary, and will thus depend on the uncertainty in critical dry diameter.

## 4   Results for Laboratory Aerosol

The AAC-CCNC measurements of sucrose were reported at varying CCNC instrument supersaturations (0.2 to 0.6%). The aerosol particles classified with the AAC were counted using a CPC and CCNC. An example dataset of CN and CCN number size distributions measured at 0.39% supersaturation is shown in Figure 4(a). The CN and CCN particle counts are plotted

against the aerodynamic diameters. Error bars in the y-axis denote the relative uncertainties in the CN and CCN number concentrations. The errors in CN and CCN concentrations are calculated from counting uncertainties of the CPC and CCNC, and uncertainties in the instrument flow rate of CPC and CCNC. Details of uncertainty estimation for the CN and CCN counts are provided in Moore et al. (2010) and are briefly described in supplemental section S3. The observed relative uncertainties in the CN and CCN concentrations were <1% for every aerodynamic diameter, which indicates that counts are repeatable.

Figure 4(b) shows the size-resolved activation ratio ($R_a = \frac{CCN(S,D_p)}{CN(D_p)}$) for 0.39% supersaturation ($S$), where $CCN(S,D_p)$ is the CCN measurement at the constant $S$ and $D_p$ divided by CN measurements at constant $D_p$, $CN(D_p)$. The sigmoidal fit applied to the $R_a$ is also shown in Figure 4(b). Error bars on the y-axis in Figure 4(b) show the uncertainties in $R_a$. The $R_a$





uncertainties were calculated by propagating the uncertainties in CN and CCN number concentrations. Details of the estimation of y-axis uncertainties of $R_a$ in Figure 4(b) are also provided in supplemental section S3.

In both Figure 4(a) and 4(b), the error bars along the x-axis show uncertainties in aerodynamic diameters estimated using the AAC TF. The x-axis uncertainties in Figure 4(a) and 4(b) decreased with increasing aerodynamic diameters. The decrease in the x-axis uncertainties can be explained using Figure 5(a) which shows the AAC TF for non-ideal particle behavior. For non-ideal AAC TF, the increase in the AAC resolution can be attributed to a monotonic increase in the transmission efficiency ($\lambda_\Omega$) and transfer function width factor ($\mu_\Omega$) with respect to the aerodynamic diameter (Figure 5(b)). Figure 5(a) shows that

the AAC TF broadening decreases with an increase in aerodynamic diameter for a fixed sheath flow rate. This is likely due to a reduced classifier flow effect with increasing aerodynamic diameter (Johnson et al. (2018)). As a result, the AAC resolution increases with increasing aerodynamic diameter. An increased resolution results in a decrease in the x-axis uncertainty with increasing particle sizes. In other words, the diffusion losses decrease with an increase in the mobility mass and aerodynamic diameter, in turn decreasing $R_a$ uncertainties associated with AAC particle size-selection and counting. From our AAC-CCNC

measurements at 8 L min$^{-1}$, the minimum AAC resolution was 10.1 to prevent excess transfer function broadening, and improve the accuracy of the size-resolved measurements. Figure 5(a) is a direct result of Eq. 13 and suggests that reducing the error in size measurement reduces the magnitude of error propagation for single-hygroscopicity parameter ($\kappa$).

    The critical dry aerodynamic diameter at 0.39% supersaturation was approximately 130nm. The AAC-CCNC sigmoidal fitting is similar to that applied by SMCA (Moore et al. (2010)). However, the sigmoid applied to the AAC-CCNC measure-

ments does not require the correction of multiple charging artefacts. The critical dry aerodynamic diameter (130nm) and $S$ (0.39%) were then combined using the Köhler theory framework (Section 2.2) to estimate the single-hygroscopicity parameter ($\kappa$) of sucrose. At 0.39% the $\kappa$ was found to be 0.041 (Eq. 1). This had a ~51% difference with respect to the theoretical $\kappa$ (0.084) of sucrose. Like Figure 4(b), Figure 6 shows the size-resolved activation ratios estimated from the measured number size distributions at 5 different supersaturations (0.23%, 0.31%, 0.39%, 0.48%, and 0.57%). The uncertainties associated with

the aerodynamic diameters and their corresponding activation ratios are also shown on the plot. In addition to this, the critical dry aerodynamic diameters obtained from the size-resolved activation ratios at respective supersaturations are provided. For every set of size-resolved activation data, the y-axis uncertainties increase, while the x-axis uncertainties decrease with increasing aerodynamic diameters (Table 1). $\kappa$ was calculated for each supersaturation using Eq. 1, and the propagated uncertainty from the critical dry volume equivalent diameter was calculated using Eq. 16. The $\kappa$ and associated uncertainties were averaged for

5 sets of measurements at every instrument supersaturation. Accounting for changing instrument supersaturations, the mean $\kappa$ for the set of 5 aerodynamic measurements was 0.036±0.008 (slightly < 0.041, the mean $\kappa$ at 0.39% supersaturation). The 0.036 $\kappa$ value was less than previously reported sucrose $\kappa$ values from electrical mobility CCN measurements (in the range of 0.06-0.1) (Xu et al. (2014), Ruehl et al. (2016), Dawson et al. (2020)), as well as the theoretical sucrose $\kappa$ (0.084). This relatively large differences between $\kappa$ values are attributed to the use of the aerodynamic diameter in Eq. 1.

Aerodynamic diameters are generally overpredicted as they are based on a spherical particle with a density ($\rho_0$) = 1000kg/m$^3$, and is likely true in the case of sucrose as its bulk density = 1586kg/m$^3$ (Guard (1999)), which is significantly larger than $\rho_0$. In such a case, a more reliable measure of particle size is required to improve the accuracy of the AAC-CCN



hygroscopicity estimates. The measured aerodynamic diameters were converted into volume equivalent diameters by accounting for the particle dynamic shape factor and true particle density (Tavakoli and Olfert (2014)). Size-resolved shape factor

measurements of sucrose are described in detail in the supplemental section S4. Dynamic shape factor ($\chi$) = 1 corresponds to spherical particles, and $\chi$>1 marks a deviation of particle shape from sphericity. For sucrose particles with aerodynamic diameters between ~100nm and 250nm, the size-resolved dynamic shape factor was approximately 1 and was observed to be only as high as ~1.1 for particles with $d_{ae}$=100nm. Table 1 provides a summary of critical dry aerodynamic diameters and their volume equivalent counterparts found in this study for CCN measurements at different supersaturations.

The $\kappa$ computed using Köhler theory from 5 different dry aerodynamic activation diameters and their respective volume equivalent diameters are summarized in Table 1. Physically, the volume equivalent diameter represents a spherical particle with the same mass as that of a non-spherical aerodynamic particle. However, the volume equivalent diameter accounts for the aerosol density as well as the deviation of the aerosol particles' shape from sphericity and improves the accuracy of hygroscopicity estimates. The mean sucrose $\kappa$ computed from critical dry volume equivalent diameters was estimated to be

0.09±0.006. The critical dry volume equivalent diameters combined with their respective critical supersaturations provided estimates of sucrose $\kappa$ that are in a better agreement with the theoretical and previously reported hygroscopicity values.

## 5   Summary, Recommendations and Implications

This study presents the AAC-CCNC experimental setup. The presented methodology can be applied for CCN activity analysis of different aerosol species. Aerosol size-selection with the AAC does not require charging of particles; thus the AAC-CCNC

coupling generates truer monodisperse aerosols, ideally sigmoidal activation data, and improves the accuracy of size-resolved measurements. For AAC-derived critical dry aerodynamic diameters, the sizing uncertainty is larger at low particle sizes and reduces with an increase in particle size (Table 1). Thus, larger critical dry aerodynamic diameters are preferred with the AAC-CCNC setup and so the AAC-CCNC setup more applicable for CCN measurements of low-hygroscopicity aerosols. It should be noted that this phenomenon is reversed for electrical mobility measurements. In the DMA, this can be attributed to increased

diffusion losses due to a drop in transmission efficiency for the particles larger than 100nm. A similar observation can be made based on the findings in Figure 5 of Johnson et al. (2018). To reiterate, the uncertainties in the electrical mobility diameter increase for larger particle sizes (Figure 7). Table S5.1 provides the measure of uncertainties in aerodynamic and mobility diameters of sucrose at the same supersaturations.

An optimum range of aerodynamic diameters for CCN measurements can be suggested based on the findings of this work.

There are fewer particles larger than 0.5$\mu$m generated via Collison atomization. Therefore, atomization produces extremely low number concentrations for particles larger than 0.5$\mu$m, which can significantly reduce the counting statistics. This suggests that ~0.5$\mu$m was a suitable upper limit of aerodynamic diameters for laboratory AAC-CCNC measurements. The lower size limit can be defined using the AAC resolution, the TF broadening and hence the flow rates used in the experiments. The sample and sheath flow rates were set to 0.8 L min$^{-1}$ and 8 L min$^{-1}$, respectively. Additionally, AAC TF equations (section 3), indicate

a lower size limit of ~85nm. The minimum measurement resolution to obtain good counting statistics corresponding to any



aerodynamic diameter 85nm was 10.1. Based on the TF analysis in this paper and the previously described CCN activity measurements (Rose et al. (2008), Moore et al. (2010)), it can be inferred that AAC is useful for particle classification and size-resolved measurements for relatively larger particles in the submicron regime. Furthermore, 85nm is a reasonable lower limit for CCN measurements of low-hygroscopicity aerosols. Low-hygroscopicity aerosols (predominantly organics with $\kappa \leq 0.2$;

Petters and Kreidenweis (2007), Xu et al. (2014), Vu et al. (2019), Wang et al. (2019)) do not activate readily at smaller particle sizes and atmospherically relevant supersaturations (<1%). The laboratory number size distribution measurements for such aerosols are reliable at low to mid-range supersaturations with the AAC-CCNC setup.

The uncertainty analysis in this work shows that size-resolved aerodynamic measurements are precise. However, the accuracy of the aerosol hygroscopicity estimates from aerodynamic measurements is low; this is seen from the lack of agree-
ment between aerodynamic diameter-derived $\kappa$ and previously reported well-accepted $\kappa$ values of sucrose. The aerodynamic diameter can be converted to a volume equivalent diameter if an additional aerosol classifier is used in series with the AAC (Yao et al. (2020)). We measured size-resolved dynamic shape factor ($\chi$) with a DMA-AAC setup to convert the aerodynamic diameters to their respective volume equivalent diameters. The volume equivalent diameter of the particles were estimated by incorporating $\chi$ and known aerosol density (Eq. 14). The aerosol hygroscopicity estimates using volume equivalent diameters
in the analysis showed good agreement with previously reported sucrose hygroscopicity values.

Overall, the AAC-CCNC coupling offers a promising tool for obtaining size-resolved CCN activity measurements for challenging low-hygroscopicity organic aerosols. Using the AAC-CCNC setup, the measurements and activation properties are obtained in terms of aerodynamic diameter. However, the sole use of aerodynamic diameters should be avoided in the context of CCN activity. CCN activity depends on particle size and chemistry; aerodynamic diameters assume a constant
density of 1000kg/m$^3$, therefore neglecting the densities of different chemical species. The use of aerodynamic diameters for CCN analysis has significant consequences for the representation of aerosols and for the estimation of hygroscopicity ($\kappa$). Future work should add the dynamic shape factor and particle density in aerodynamic diameter-derived CCN activity analysis. The additionally known parameters improve agreement between the measured and theoretical $\kappa$ values.

*Code and data availability.* The PyCAT code is available for public use through GitHub - https://github.com/kgohil27/PyCAT-1.0. The mea-
surement data can be provided by the authors on request.

*Author contributions.* KG designed the analysis for the AAC-CCN experimental data. AAA conceived the idea for the study; designed and developed the experimental methodology. Both authors contributed to the writing and preparation of the manuscript.

*Competing interests.* The authors declare that they have no conflict of interest.





*Acknowledgements.* This material is based upon work supported by the National Science Foundation under Grant No. NSF: CHEM-1708337.



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





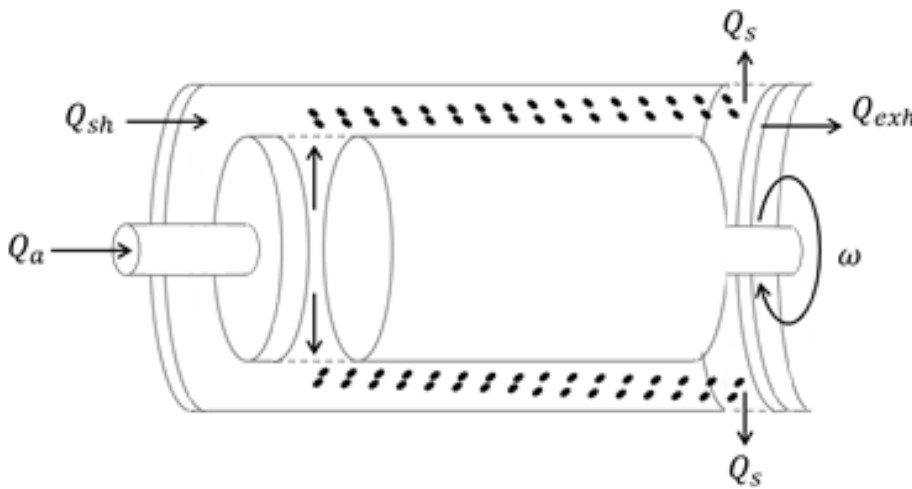

**Fig. 1.** The schematic of how the Aerodynamic Aerosol Classifier (AAC) size-selects particles. The rotating 2-cylinder arrangement of the AAC subjects particles to a centrifugal force that is balanced by the drag force along the axial direction.





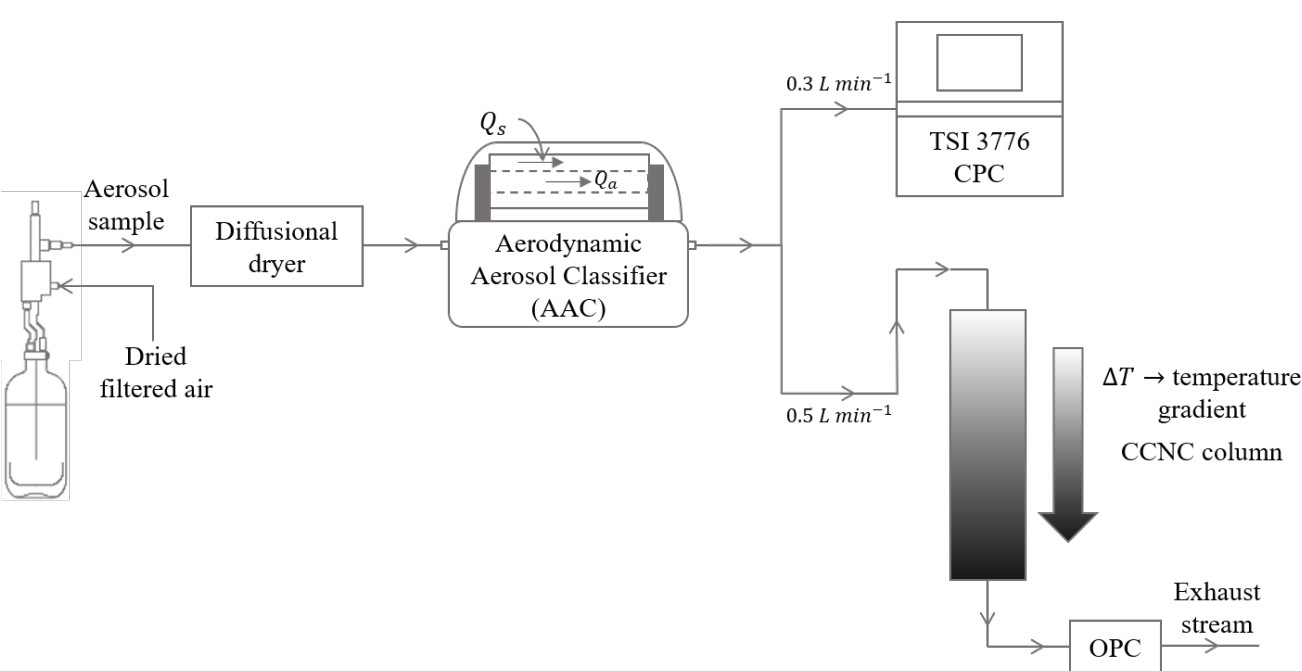

**Fig. 2.** AAC-CCNC experimentation setup for measuring the size-resolved number concentration of aerosols.



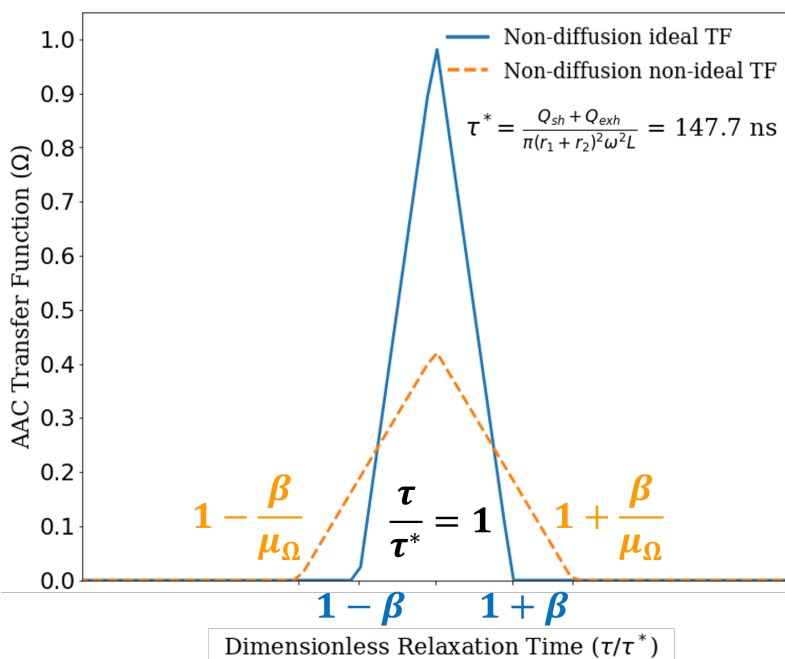

**Fig. 3.** The ideal (blue) and non-ideal (NI) (orange) AAC transfer functions based on the particle streamline non-diffusion (ND) model as developed by Tavakoli and Olfert (2013). The transfer functions are shown relative to 150 nm aerodynamic diameter as the setpoint. This corresponds to a relaxation time setpoint of 147.7 ns. It can be observed that the NI transfer function maximum is significantly reduced as compared to the ideal transfer function which is attributed to a reduced transmission efficiency for the NI transfer function. Additionally, the transfer function broadening is higher for NI transfer function which is quantified using the transfer function width factor. Overall, the NI transfer function provides an improved basis for particle size-selection using the AAC.





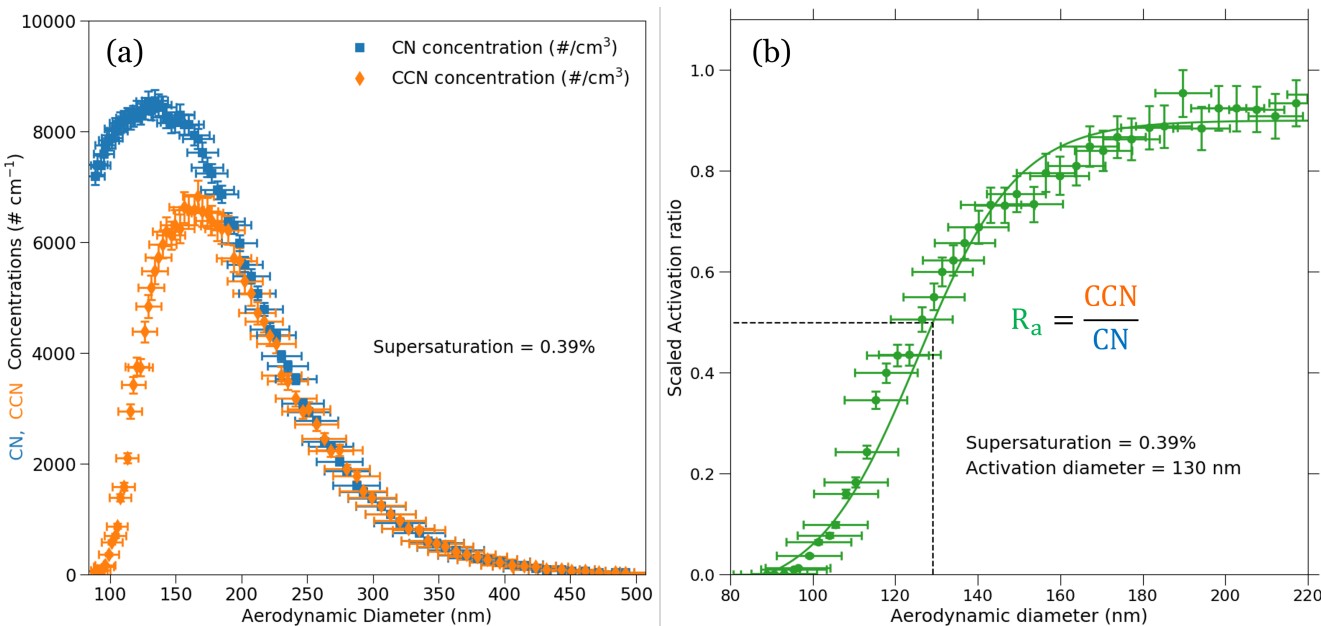

**Fig. 4.** The CN and CCN number size distributions (a) and the corresponding size-resolved activation ratio (b) for sucrose are shown. The measurements was performed at a supersaturation of 0.39%. The activation aerodynamic diameter was found to be about 130 nm from the activation ratio obtained using the size-resolved measurements. The dry aerodynamic activation diameter corresponds to the 50% activation efficiency which has been denoted with the set of dotted black lines. Furthermore, the uncertainties in the aerodynamic diameters, CN and CCN number concentrations and size-resolved activation ratio are also denoted on the plot using their respective error bars.





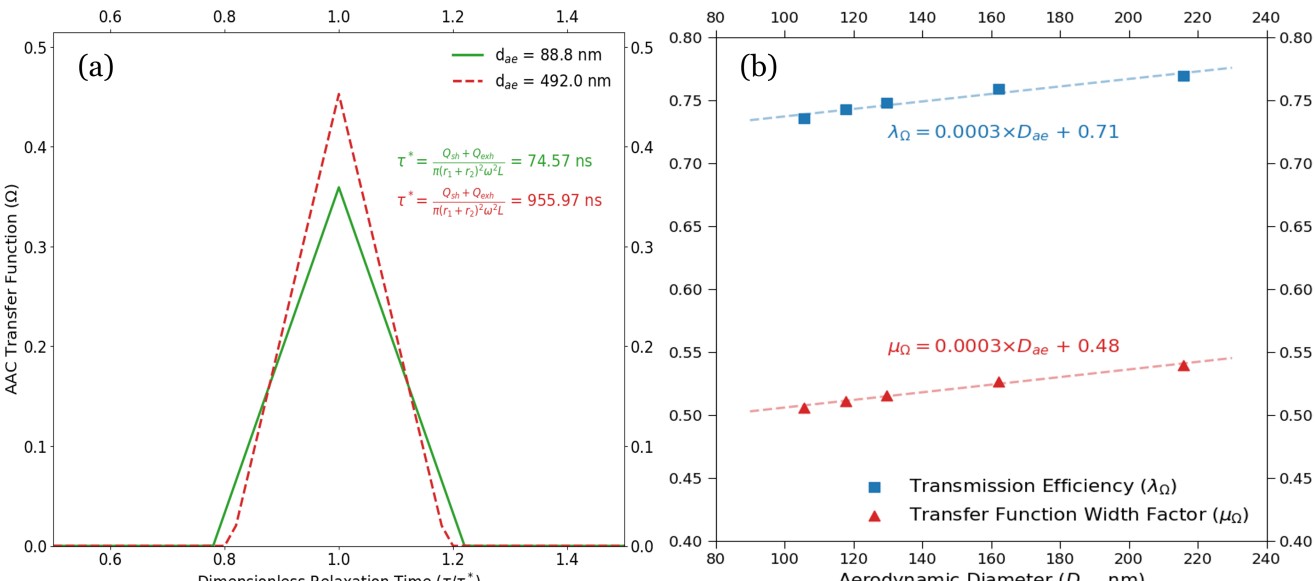

**Fig. 5.** (a). ND-B-NI transfer functions for low-flow measurement conditions from the AAC-based setup are shown. The transfer functions are plotted with respect to dimensionless relaxation time. 90 nm and 392.3 nm were respectively the lower and upper aerodynamic diameter limits for the measurements. The corresponding relaxation times are highlighted for the aerodynamic diameter setpoints. The resolution increases and hence the measurement uncertainties decrease with an increase in the particle aerodynamic diameter. (b). The size-dependent transmission efficiency ($\lambda_\Omega$, blue) and transfer function width factor ($\mu_\Omega$, red) are shown above. The marked points on the plot correspond to $\lambda_\Omega$ and $\mu_\Omega$ computed at the dry activation diameters at the set instrument supersaturations used in this study (between $0.2 - 0.5\%$). As a general trend, both the transfer function parameters increase with the increase in aerodynamic diameter. This results in an increase in the AAC transfer function resolution and a decrease in the size-related uncertainty with an increase in aerodynamic diameter (that is, the particle relaxation time). The plot also shows that the transfer function width factor is slightly more sensitive to the increase in the aerodynamic diameter, which can be followed by comparing the slopes of the linear fits of the transmission efficiency and width factor relative to the aerodynamic diameter.

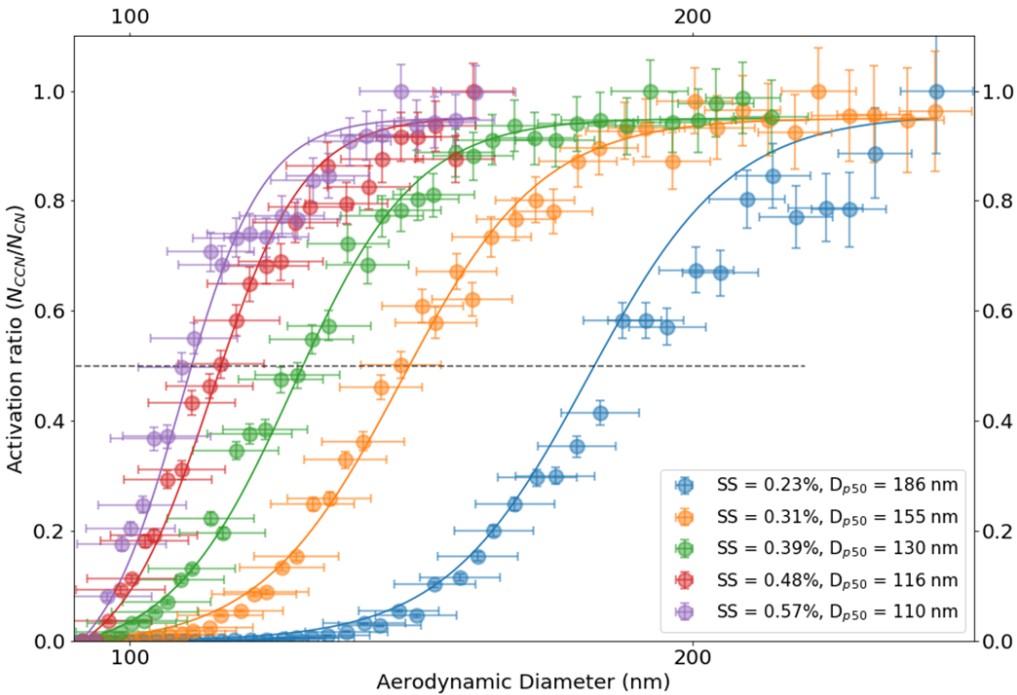

**Fig. 6.** Size-resolved activation ratio are shown over a range of instrument supersaturations as presented on the plot. Their corresponding dry activation diameters are also depicted on the plots. The dotted line passing through the 50% activation efficiency point on the plot intersects the activation ratio plots at their respective dry activation diameters. The dry activation diameter systematically decreases with increasing ambient supersaturation.



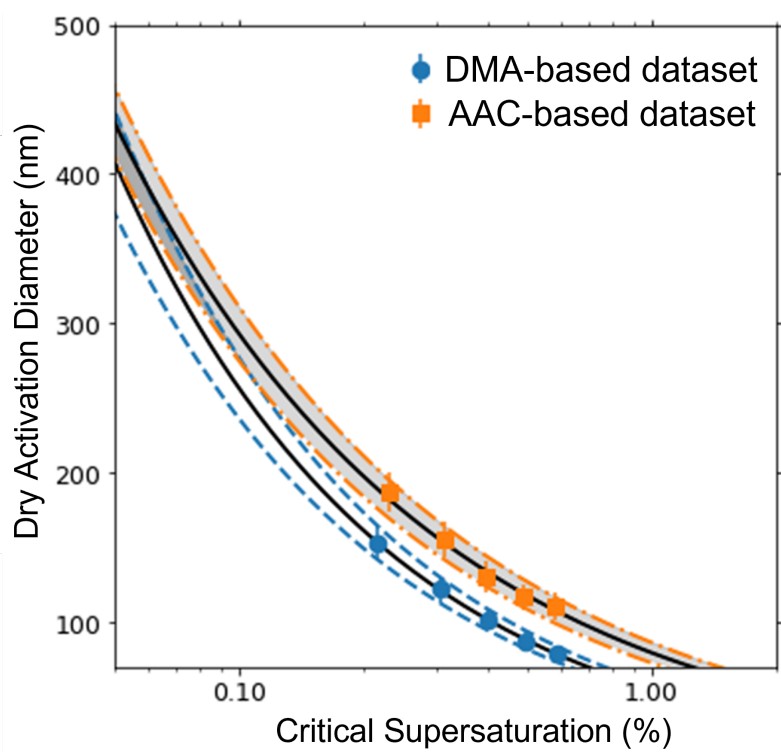

**Fig. 7.** The variation in uncertainty of size-resolved measurements using a DMA and AAC are compared in this plot. The set of orange dot-dashed lines denote the range of uncertainty in measurements in an AAC, and the blue dashed lines denote the range of uncertainty in measurements in a DMA. The black solid lines are the best fits for the size-resolved measurements for sucrose obtained using the Köhler theory.



Table 1. The table provides the analysis summary of the set of measurements performed for sucrose with the help of the AAC-
CCNC setup. At low supersaturations, sucrose has large dry activation diameters for which the measurement uncertainties are
slightly lower. Furthermore, the CCN activity predictions in terms of $\kappa$ using the Köhler theory are also accurate. With an
increase in the supersaturation the dry activation diameter reduces, and correspondingly the variations in $\kappa$ continue to rise,
being as high as about 35% at 0.58% instrument supersaturation. The conversion of dry aerodynamic activation diameters of

sucrose to their corresponding volume equivalent diameters was done with the help of dynamic shape factor. The shape factor
measurements and analysis was performed following the procedure described in Tavakoli and Olfert (2014).



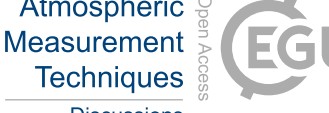

**Table. 1.** Summary of Sucrose Measurements and Shape Factor Corrections.

| Instrument Supersaturation (S) | $D_{p50}$ (nm) | Uncertainty $D_{p50}$ | $\kappa^{*}$ | Relative Difference - Measured $\kappa$ v/s $\kappa_{theoretical}$ & | Dynamic Shape Factor ($\chi$) | Volume Equivalent Diameter $D_{v,p50}$ (nm) $^{\mathcal{L}}$ | $\kappa^{e}_{corrected}$ | Relative Difference $\kappa_{corrected}$ v/s $\kappa_{theoretical}$ |
|---|---|---|---|---|---|---|---|---|
| 0.23% | 186 | ±7% | 0.04 ± 0.0083 | 52.4% | 1.019 | 140 | 0.093 | 10.7% |
| 0.31% | 155 | ±7.3% | 0.038±0.0082 | 54.8% | 1.023 | 116 | 0.087 | 3.6% |
| 0.39% | 130 | ±7.6% | 0.041±0.0092 | 51.2% | 1.044 | 95 | 0.099 | 17.8% |
| 0.48% | 116 | ±7.8% | 0.04 ± 0.0091 | 52.4% | 1.037 | 86 | 0.088 | 4.76% |
| 0.57% | 110 | ±8% | 0.032±0.0074 | 61.9% | 1.052 | 80 | 0.081 | 3.6% |

\* - $\kappa$ determined using the dry aerodynamic activation diameter in the Köhler theory framework
& - theoretical $\kappa$ of sucrose determined from ideal Köhler theory = 0.084
$\mathcal{L}$ - volume equivalent diameter including the dynamic shape factor with the dry aerodynamic activation diameter
∈ - $\kappa$ determined using the volume equivalent activation diameter in the Köhler theory framework