# Peer review of "Cloud Condensation Nuclei (CCN) Activity Analysis of Low-hygroscopicity Aerosols Using the Aerodynamic Aerosol Classifier (AAC)"

_Atmospheric Measurement Techniques, 2021_

## Author Comment (AC1)

We thank the reviewers for their insightful comments and feedback. We have provided our point by point responses in blue text.

**Cloud Condensation Nuclei (CCN) Activity Analysis of Low-hygroscopicity Aerosols Using the Aerodynamic Aerosol Classifier (AAC)**

**Responses to Referees**

**Referee 1:**

This study describes the use of a relatively new system to size-select particles known as the Aerodynamic Aerosol Classifier (AAC) manufactured by Cambustion to measure cloud condensation nuclei (CCN) activity of aerosols. This combination of AAC + CCN counter has not yet been characterized in the literature and the study uses a single model compound, sucrose, to verify the theoretical calculations and uncertainties calculated from the transfer function of the AAC. This work offers a nice extension of the work by Moore et al. for the differential mobility analyzer + CCN counter system and should provide a useful tool for future users as the system becomes more popular. As part of the study, a Python package is available through GitHub for the activation diameter and uncertainty calculations. Although I checked that it was available, I did not run the package myself and cannot comment on its capabilities. In addition, I was only able to follow the equations at a high level and have some specific comments listed below.

Comment 1: I see four minor issues associated with this study. The foremost is the use of the aerodynamic diameter to initially calculate the hygroscopicity parameter. The main reason that the diameter is in the original Kohler equation is to determine the number of soluble moles in the particle. It would therefore make sense to present the calculations using the volume-equivalent diameter first, especially since the point of this study is to demonstrate the usefulness of the AAC and to present the results in the best possible light. I could see keeping the discussion about the use of aerodynamic diameter when the particle density and shape factor are unknown since this will be the case for some applications. I realize that this point is entirely stylistic and I leave it to the editor and the authors to determine whether this suggestion should be implemented. Related to this point, the tone of the article would be improved if lines 328-330 of the conclusions were reframed to say that it is important to use the volume-equivalent diameter when possible to calculate kappa, since that is more representative of the terms in the Kohler equation, instead of framing it around the aerodynamic diameter.

Response: This is correct. The diameter in the Köhler equation is used to determine the soluble moles and is always best represented by the volume-equivalent diameter. However, much of the existing CCN literature that uses DMA measurements does little to no conversion to volume equivalent diameters. It is thus often accepted that the electrical mobility diameter is an okay and valid substitute into droplet equations. Thus, the reason why we chose to present the results using the aerodynamic diameter was to draw an analogy between AAC and DMA-based CCN measurements. Furthermore, a somewhat surprising find is that generally spherical

aerosols, their electrical mobility diameters can be directly used in the Kohler equation. This can be clearly understood from the following equation -

$$\frac{C_c(d_{ve})}{d_{ve}\chi} = \frac{C_c(d_{mo})}{d_{mo}}$$

In the above equation, Cc is the size-dependent Cunningham's slip correction factor, \chi is the dynamic shape factor, dmo is the measured electrical mobility diameter and dve is the volume equivalent diameter. If Cc(dve) \approx Cc(dmo) and \chi \approx 1, then dve \approx dmo. Furthermore, high quality hygroscopicity predictions with low uncertainties can be made without the conversion from electrical mobility diameters to volume equivalent diameters.

This led us to test the applicability of the aerodynamic diameters for obtaining the hygroscopic properties. We found that even though the aerosol may be composed of nearly spherical particles, there can be uncertainties in hygroscopicity calculations from aerodynamic diameters. These uncertainties result from differences in the aerosol density with respect to the reference density of 1000 kg m-3. This can be clearly seen from the following equation -

$$d_{ae} = d_{ve}\sqrt{\frac{\rho_p C_c(d_{ve})}{\chi \rho_0 C_c(d_{ae})}}$$

where \rho_p is the aerosol density, \rho_0 is the reference density and dae is the measured aerodynamic diameter. Even if \chi \approx 1 for an aerosol (such as, sucrose), the difference between \rho_p and \rho_0 could be large and contribute to large differences between dae and corresponding dve. Moreover, understanding the differences between the diameters provides a background for how the AAC can be used to obtain the dynamic shape factor of aerosols. In this revision of the manuscript, we have endeavored to add more of this understanding. The changes are not specific to one section and we have highlighted this concept throughout the revised manuscript.

Comment 2: A more important issue is the relative uncertainties presented in equations 12, 15 and 16. Errors are never subtracted, otherwise a large uncertainty in the terms in the denominator could reduce the overall relative uncertainties, which is unreasonable. There should be an absolute value around the terms, leading to the relative errors being summed (i.e. all the subtraction signs should be changed to addition). Can the authors explain why they are adding the relative errors instead of adding them in quadrature? Once corrected, the new uncertainties should be propagated through the rest of the manuscript in the revised version.

Response:

Paper:

$$\frac{\Delta\tau}{\tau} = \frac{\delta Q_{sh}}{Q_{sh}} - 2\frac{\delta\omega}{\omega} - 2\frac{\delta r}{r} - \frac{\delta L}{L} \qquad (12)$$

$$\frac{\Delta d_{ve}}{d_{ve}} = \frac{\Delta d_{ae}}{d_{ae}} + \frac{1}{2\chi\rho_0}\frac{\Delta C_c(d_{ae})}{C_c(d_{ae})} - \frac{1}{2\rho_p}\frac{\Delta C_c(d_{ve})}{C_c(d_{ve})} + \frac{1}{2}\frac{\Delta\chi}{\chi} \qquad (15)$$

$$\frac{\Delta\kappa}{\kappa} = -3 \cdot \frac{\Delta D_{p50}}{D_{p50}} \qquad (16)$$

The reviewer is correct and the uncertainties were not subtracted. The former equations of uncertainty (shown above) were only used to qualitatively represent the total uncertainties in the relaxation time and volume equivalent diameter resulting from the respective partial derivatives. The true uncertainties in the paper were evaluated by taking the square root of the sum of squares of individual partial derivatives.

The above expressions have been changed in the revised manuscript as follows to reflect the actual calculations presented following:

$$\left[\frac{\Delta\tau}{\tau}\right]^2 = \left[\frac{\delta Q_{sh}}{Q_{sh}}\right]^2 + 4\left[\frac{\delta\omega}{\omega}\right]^2 + 4\left[\frac{\delta r}{r}\right]^2 + \left[\frac{\delta L}{L}\right]^2 \qquad (12)$$

$$\left[\frac{\Delta d_{ve}}{d_{ve}}\right]^2 = \left[\frac{\Delta d_{ae}}{d_{ae}}\right]^2 + \frac{1}{4}\left[\frac{1}{\chi\rho_0}\frac{\Delta C_c(d_{ae})}{C_c(d_{ae})}\right]^2 + \frac{1}{4}\left[\frac{1}{\rho_p}\frac{\Delta C_c(d_{ve})}{C_c(d_{ve})}\right]^2 + \frac{1}{4}\left[\frac{\Delta\chi}{\chi}\right]^2 \qquad (15)$$

$$\frac{\Delta\kappa}{\kappa} = 3 \cdot \frac{\Delta D_{p50}}{D_{p50}} \qquad (16)$$

Comment 3: Another concern is that the uncertainty in the calibrated supersaturation of the CCN counter is not included in the overall uncertainty calculations. The authors should include a discussion of the uncertainties in the diameters measured by the DMA used in the calibrations, the uncertainty of the fitted critical diameter (as shown in the Supplement), and their effect on the calibrated supersaturation. This should then be included in the uncertainty in kappa.

Response: In the original manuscript, we tried to focus the uncertainty discussion on the effects of particle diameter measurement and not necessarily the CCN counter. In the revised manuscript, we have incorporated the reviewers' concerns regarding the uncertainty in

calibrated supersaturation with the DMA with additional discussion and reference to calculations and extensive work published by Rose et al (2008), and Roberts and Nenes et al (2005). The following statements have been added to the text in the "Summary, Recommendations and Implications" section:

"It should be noted that the uncertainty calculations presented in this manuscript solely focus on the uncertainties from changing sizing instrumentation used before the CCN counter. That is, if one uses the same CCN counter, the uncertainty in the supersaturation (from changes in the delta T, flow rate, and delta P) are constant. One can add additional calculations of error in supersaturations by referring to Roberts and Nenes (2005) and Rose et al (2008). If the user intends to perform CCN measurements using the AAC and DMA, they should run the CCN measurements at the same time."

Comment 4: A final concern is that the final uncertainty for the measured kappa value presented on line 290 is 0.006. Was this calculated from Equation 16? This value is comparable to the standard deviation of the kappas presented in Table 1 (0.007) and suggests that the repeatability of the measurement, over a range of supersaturations, is worse than the instrumental uncertainties and all the analysis presented. Was this true of repeated measurements at the same supersaturation? Please provide some perspective on this.

Response: The kappa 0,09 +/- 0.006 is calculated from Equation (1), and is the value obtained from aerodynamic values corrected with shape factor and particle density. The associated uncertainty was calculated for every set of measurements using Equation (16). The uncertainty is derived from measurements in Table 1 and thus it makes sense that they have comparable uncertainty. Indeed, the uncertainty is supersaturation dependent; as the uncertainty changes with particle sizes, and particle size uncertainty is a function of the transfer function of the instrument used. Table 1 also shows the uncertainty at each given supersaturation. The uncertainty is affected by both uncertainty in particle measurement and CCN measurement, not just particle measurement itself. Indeed, there are higher uncertainties with supersaturation at lower supersaturations (with smaller delta T's). The significant digits in the kappa uncertainties were observed to be nearly the same - 0.0061 v/s 0.0065 - for aerodynamic diameters and volume equivalent diameters, respectively.

**Referee 2:**

In this manuscript the authors are characterizing a measurement system combining the Aerodynamic Aerosol Classifier (AAC) and the DMT CCN counter in order to measure the CCN efficiency of size selected aerosol particles. The purpose is to determine the uncertainty of the AAC classification and propagate the error to the kappa-values determined from the 50% activated fraction of aerosol in the CCN counter. The authors also compare the obtained kappa-values to those measured using a conventional DMA-CCNC system.

The manuscript is well within the scope of AMT, and the measurements appear sound, but the uncertainty analysis could be improved. I have a few comments/questions related mostly to the activated fraction, the sigmoid function, and their impact on the uncertainty of the obtained kappa.

Comment 1: First, the sigmoid curves of the size-resolved activated fractions measured using the AAC-CCNC are much wider than those measured with the DMA-CCNC. This can be seen e.g. by comparing the curve of Fig. 4(b) to the green (?) curve of Fig. S3 (note that there are 7 curves in the latter figure but only 6 rows in Table S2 so it is not completely clear which curve corresponds to which row). What are the reasons for the wider sigmoid? Does it follow directly from wider transfer functions of AAC compared to DMA? Would not a wide sigmoid in itself impact the uncertainty of kappa via making the diameter of 50% activated fraction more uncertain?

Response: Figure 4(b) shows the activation curve with respect to the aerodynamic diameters. The use of the aerodynamic diameters in this plot is the main reason for the wide sigmoidal fit. We believe that the morphological and density differences in the aerodynamic diameters of the particles compared to their volume equivalent diameters cause the sigmoidal broadening. Furthermore, if the activation ratios are plotted with respect to the volume equivalent diameters, we obtain a narrower sigmoidal fit.

Comment 2: Secondly, the fitted sigmoid curves do not appear to reach unity but seem to approach a constant value of something like 0.95. Is this true? (Please provide the sigmoid fitting functions in the supplement.) If it is true, does it mean that about 5% of the particles are lost in the CCN counter? Wouldn't it then be logical to determine the critical diameter from 50% of the maximum activated fraction of the fitted sigmoid curve and not from 50% of the input aerosol concentration? (This obviously applies to the DMA-CCNC measurements as well).

Response:
The raw data for this figure has a maximum at about 1.06 and the sigmoidal fit applied to this data plateaus at about 0.99. The critical diameter is defined at the inflection point corresponding to the 50% max efficiency (0.496 for the measurements shown in Figure 4). The overall figure (initially between 0 and 1.06) was normalized between 0 to 1 after performing the calculations for sigmoidal fitting. Hence, the sigmoidal fit in the figure as shown seems to plateau at 0.96

The figure was generated after the fit was applied and skewed the efficiency in the plots. We have revised such that the black dashed lines are correctly passing through the 50% activation efficiency in the new plots in Figures 4(b) and 6.

Equations for sigmoid fitting functions have also now been added to the supplement. It should be noted that the CPC and CCN counters are separate instruments with different optical counting efficiencies. Thus it is common for the two separate instruments (even of the same brand) to be within 10% counting efficiency. It is a common practice to normalize the data so that the activation ratio plateaus at 1.

Comment 3: Finally, there obviously is some statistical uncertainty in the fitted sigmoid curves, For example, at high activated fractions, the blue datapoints in Fig. 6 are rather scattered and mostly below the sigmoid. Can you determine what is the error of the critical diameter associated with the statistical uncertainty of the fitting function, and how it further impacts the error estimate of the resulting kappa value? Or is the statistical uncertainty perhaps within the error limits caused by the uncertainties of the measured aerodynamic diameters?

Response: The uncertainties in the sigmoidal fitting were not explicitly shown in the activation plots. However, the uncertainties in kappa estimates resulting from the uncertainties in the aerodynamic diameters or volume equivalent diameters (depending on the analysis) were calculated using the following expression -

$$\frac{\Delta \kappa}{\kappa} = -3 \cdot \frac{\Delta D_{p50}}{D_{p50}}$$

Furthermore, the scattering of the activation ratio points in Figure 6 is mostly seen at larger particle sizes. This is attributed to the reduced counting statistics at larger sizes in the particle size distribution. Lower counting statistics are more likely to be found at higher sizes for which particle number concentrations can be low (<50 #/cc), causing the activation ratio to fluctuate around an average. The average is usually around the plateau of the fitted sigmoid curve. Hence, the likelihood of retaining any uncertainties from reduced particle counts in subsequent calculations is low.